# The Impact of the Platform Economy on Urban–Rural Integration Development: Evidence from China

Wang Zhang

School of Economics and Management, Northwest University, Xi'an 710127, China; zhangwang@nwu.edu.cn

**Abstract:** As a new economic system composed of data-driven, platform-supported, and network-coordinated economic activity units, the platform economy provides a new impetus for the integrated development of urban and rural areas. This paper uses China's provincial panel data from 2011–2021 to measure the development level of the platform economy and the urban–rural integration development index using the entropy weight method and empirically tests the direct and indirect effects of the platform economy on urban–rural integration development. The research finds that (1) the platform economy has a significant role in promoting the development of urban–rural integration. After introducing a series of robustness tests such as instrumental variables estimation, the results are still significant. (2) Regional innovation capability is an important transmission mechanism for the platform economy to enable urban–rural integrated development. The platform economy can indirectly promote urban–rural integrated development by improving regional innovation capability. (3) There is a significant mesomeric effect between the traditional factors of production and the platform economy and the mesomeric effect of capital, technology, land, and labor factors increased in turn. (4) The digital twin platform is a kind of digital land generated in the platform economy and has become an important place and value source for products in the digital era.

**Keywords:** platform economy; urban–rural integrated development; regional innovation capability; digital land





## 1. Introduction

Since the reform and opening up, China's urbanization rate has been continuously increasing and the scale of cities has steadily expanded. With the comprehensive victory of poverty alleviation, the focus of China's "agriculture, rural areas, and farmers" work has gradually shifted to comprehensively promoting rural revitalization, and urban–rural integration development has gradually become a key task in the development of urban–rural relations in China. With the development of modern information technologies such as the internet, cloud computing, and big data, the platform economy has rapidly emerged as a new business form and model, becoming an important driving force for promoting urban–rural integration development [1,2]. By guiding urban technology, network, talents, information, and other resources to flow fully to the countryside, the platform economy has weakened the geographical distance, promoted the two-way flow of urban and rural resource elements, and provided new impetus for the integrated development of urban and rural areas [3,4]. The strategic plan for rural revitalization (2018–2022) and the no. 1 central document in 2019 both regarded the "digital rural strategy" as an important carrier to improve China's urban and rural relations and build an integrated urban and rural development model. As the platform economy deepens the empowerment of all aspects of urban and rural integration, the positive effects of the positive interaction between the two become increasingly apparent. Therefore, it is of great practical significance to deeply explore the impact mechanism of the platform economy on urban and rural integrated development and discuss whether the platform economy can break the urban–rural dual

structure to solve the main contradiction of unbalanced and inadequate development in China.

In resolving these issues, many scholars, such as Cardona [5], Miyazaki [6], Mouelhi [7], and Forero [8], have conducted a large number of studies on this area, mainly focusing on the following aspects. First, the platform economy can influence economic and social operations by accelerating capital accumulation, optimizing factor allocation, and other forms, which play an important role in promoting the integrated development of urban and rural areas and achieving the goal of coordinated development of urban and rural areas [9,10]. The platform economy solves the problem of information constraints and cognitive constraints in traditional economic society by means of "algorithm and data", in order to system integrate social resources and provide resource allocation and utilization efficiency [11]. Second, the platform economy links basic public services with basic public facilities through the connectivity effect of data elements to attract talent, labor, capital, and other elements to the countryside actively [12]. The platform economy has established a modern urban–rural relationship. The coverage of network infrastructure and the application of information technology in rural areas determine the convenience and efficiency of information communication between urban and rural areas. The efficient information communication between urban and rural areas is conducive to accelerating the diffusion of urban and rural space, promoting the employment of the rural population, and reducing the urban unemployment rate [13]. Third, the platform economy continues to empower regional innovation by promoting enterprise innovation, business model innovation, and technological innovation. Pisano et al. found that the platform economy model can promote the instantaneous matching between the supply and demand sides in traditional industries and improve the professional level and personalized and precision capabilities of traditional industries [14]. Armstrong believes that, compared with the traditional economy model, the platform economy has subversively changed the ways of human production and consumption, and has played a positive role in integrating economic resources, optimizing the industrial division of labor, and improving industrial production efficiency [15] Drahokoupil and Jepsen believed that the emergence of the platform economy also provided a large number of employment opportunities for the market, and promoted the transformation and upgrading of the industrial structure by driving consumption [16]. Paunov and Rollo believe that the platform economy promotes information dissemination among enterprises, improves technical efficiency, and, thus, improves enterprise performance. Companies with high efficiency and high learning ability are more likely to obtain innovation benefits from the internet [17]. Fourth, the effective measures of the role of the platform economy in promoting urban–rural integration development mainly include two aspects: digital village and platform economy in promoting rural revitalization, and quantitative research is carried out using the mesomeric effect model, spatial Durbin model, coupling analysis, threshold analysis, etc. Li found that the impact of China's platform economy on rural development shows an "inverted U-shaped" trend, which has a promoting effect on narrowing the gap between urban and rural areas [18]. Tang and Xie believe that the platform economy has a significant effect on the consumption income of farmers. On the one hand, it improves the efficiency of rural mobility, optimizes the industrial structure, and, on the other hand, it improves the employment rate of farmers [19]. Paul et al. believe that there is a high degree of coupling between the platform economy and regional innovation and digital platforms can improve the efficiency of regional technological innovation by changing the composition of the elements of regional innovation [20].

With the vigorous development of the digital economy, research on urban–rural relations has begun to focus on digital factors. Many scholars have empirically analyzed the impact of the digital economy on the urban–rural income gap [18], consumption gap [21], and rural revitalization [22]. As the main representative form of the digital economy [23], the research on the integration of urban and rural development of the platform economy is mostly theoretical analysis. As Fan pointed out, fiscal policies promoting urban–rural integration development in the context of the digital economy should be people oriented [24].

Xie and Han believe that the digital economy promotes urban–rural integration and implements digital policies tailored to local conditions. Zhou pointed out that the digital economy, through digital technology, can break through the geographical boundaries between urban and rural areas, promote the two-way flow of urban and rural resources, and is a new driving force for improving the quality of urban–rural integrated development [25]. Since the research on the platform economy and urban–rural integration development is still in the exploratory stage at this stage, and the research expansion is also relatively limited, this paper will also refer to the research on the digital economy and urban–rural integration development while focusing on the platform economy and urban–rural integration development, in order to consolidate the research foundation and expand the research ideas.

## 2. Theoretical Hypothesis

The integrated development of urban and rural areas is the process of treating urban and rural areas as an organic whole, where they are resources, markets, complements, and promote each other, effectively promoting the two-way interaction between urban and rural areas, gradually promoting the balanced development of economic and social, residents' lives, infrastructure, public services, ecological environment, and other aspects between urban and rural areas and, ultimately, achieving common development and prosperity between urban and rural areas. As a new driving force for development and a new business model, the platform economy can effectively remove the barriers of the urban–rural dual structure and coordinate the integrated development of urban and rural areas.

### 2.1. The Direct Impact of the Platform Economy on Urban–Rural Integrated Development

The platform economy helps to promote the integration of urban and rural populations [2,5]. The deep-seated registered residence system problem is the main reason for the urban–rural dual structure and the development of the platform economy provides digital technology support for the removal of institutional barriers. The digital and information-based registered residence management model can facilitate the monitoring of population mobility, help to weaken the restrictions of registered residence on population mobility, and promote the integration of urban and rural populations. At the same time, the construction of digital rural areas and the implementation of rural revitalization policies have guided numerous urban technical talents to integrate into and build in rural areas, promoting the two-way flow of urban and rural populations.

The platform economy helps to promote the integration of urban and rural economies [8]. The combination of digital technology and agricultural equipment is conducive to improving the modernization level of agricultural facilities, liberating rural labor, improving agricultural production efficiency, and promoting the development of rural agriculture. At the same time, the development of digital platforms based on digital technology has spawned new formats and provided more employment and entrepreneurial opportunities for rural residents. Farmers can promote characteristic agricultural products and develop rural tourism through online live streaming. This helps to promote rural economic development, narrow the gap between urban and rural economic development, and gradually promote the integration of urban and rural economic development.

The platform economy helps to promote the integration of urban and rural society [11]. For a long time, due to the policy bias of cities, the allocation of resources between urban and rural areas has been unreasonable, resulting in a significant gap in basic public services such as medical equipment, educational resources, and social security. The digitization of the social public-service system is conducive to addressing the urban–rural gap. For example, rural residents can express their demands through digital government platforms, share high-quality medical resources in the city through online consultation, and share first-class educational resources in the city through cloud-based online courses, which is conducive to achieving a balanced development of urban–rural public services.

The platform economy helps to promote urban and rural spatial integration [12]. The platform economy, through digital technologies such as artificial intelligence, the internet, and big data, uses digital platforms to break through urban and rural space restrictions and realize real-time sharing of urban and rural information, which can reduce the problem of urban and rural information asymmetry. Digital technology can also be used to apply urban talent, technology, facilities, and other resource elements to rural areas, implementing remote control, which can weaken the geographical distance between urban and rural areas and promote spatial integration between urban and rural areas.

The platform economy helps to promote the integration of urban and rural ecology [15]. The platform economy takes digital and information as its core production factors, which has less pollution to the environment. The integration of the platform economy and traditional industry and agriculture can change the production mode of traditional industries, help promote the digital, low-carbon, and green transformation of urban and rural industries and agriculture, improve the resource conversion rate and energy utilization rate, and reduce pollutant emissions. At the same time, strengthening the financial support of digital financial inclusion for urban and rural green low-carbon industries can also help the urban and rural ecological environment and integration. Based on the above analysis, this article proposes the following assumptions.

**Hypothesis 1 (H1).** *The platform economy can directly promote the integrated development of urban and rural areas.*

*2.2. The Platform Economy Has Indirect Impact on Urban–Rural Integration Development through Regional Innovation Capability*

Regional innovation capability refers to the ability to creatively integrate innovation resources in a specific region, and then interact with the knowledge innovation system led by universities and the technology innovation system led by enterprises to transform innovation investment into new technologies and products. The platform economy has many advantages such as resource sharing across time and space, economies of scale, and so on, which have effectively broken the spatial barriers between regional innovation systems. The platform economy is increasingly becoming an important factor to promote the improvement of regional innovation capability [18].

The influence mechanism of the platform economy on regional innovation capability [21]. First, the platform economy can provide more convenience for enterprise innovation by improving the efficiency of resource allocation, saving innovation costs, and changing the way of thinking. Based on the internet and big data, the digital platform can quickly match the supply-and-demand sides of innovation activities, and resource docking takes a shorter and simpler time. While reducing the transaction cost of factors, it improves the allocation performance and the availability of production factors. Second, the platform economy drives the development of new business models by integrating innovative elements and reshaping user behavior, thereby improving regional innovation capability. On the one hand, the openness of the platform economy has made the business network more open and the access to knowledge resources more diversified, thus promoting more technology to be developed and utilized. On the other hand, the platform economy based on the internet can break space restrictions and realize resource sharing. Finally, the platform economy can also indirectly affect regional innovation capability through human capital, R&D funds, and other factors. The application of an internet-based platform economy to a regional innovation system will accelerate the acquisition and accumulation of human capital, increase social interaction, and promote the accumulation of social capital, thus promoting the improvement of regional innovation capability.

The impact mechanism of regional innovation capability on urban–rural integration development [24]; first, regions with high innovation capabilities also have relatively high levels of economic development. Innovative technologies will drive the transformation and upgrading of traditional industries, expand their development scale, and generate a large number of employment opportunities. In pursuit of better income, people will migrate to

cities with higher economic levels, promoting population mobility and agglomeration. At the same time, the benefits generated by urban-resource factors will feed back rural areas in other forms, Promote urban–rural interaction and integration. Secondly, innovative technologies will change the extensive development model of rural agriculture, promote the recombination of production factors, improve agricultural production efficiency, drive rural economic growth, gradually narrow the urban–rural economic gap, and promote urban–rural integration development. Finally, high-tech industries are mainly concentrated in cities with high levels of economic development. As regional innovation capabilities improve, the output of high-tech industries will also increase, which can provide technical support for the innovative development of various industries within the urban area. At the same time, high-tech industries are prone to form industrial clusters, which can promote technology spillovers. At this time, the radiation and diffusion effects of urban innovative technologies can be fully utilized to promote the transformation and upgrading of rural industries in the region, effectively driving the development of the rural economy, narrowing the urban–rural gap, and promoting urban–rural integration. Based on the above analysis, this article proposes the following assumptions.

**Hypothesis 2 (H2).** *The Platform economy can indirectly promote urban and rural integrated development through regional innovation capability.*

*2.3. The Platform Economy Indirectly Promotes Urban–Rural Integrated Development through Traditional Production Factors*

The platform economy indirectly promotes the integrated development of urban and rural areas by influencing other traditional production factors and improving the two-way flow speed of traditional production factors [25,26]. The role of the platform economy in the integrated development of urban and rural areas on other traditional production factors can be summarized into two stages: the promotion stage and the leading stage; that is, solving existing problems and promoting and leading new development, thus promoting the input of traditional production factors and improving the quality of factors. At the same time, the complementary effect of other traditional factors of production and the platform economy has further expanded the capacity of traditional factors of production, which has an indirect impact through big data analysis and application and has indirectly promoted the integrated development of urban and rural areas through the platform economy, as shown in Table 1.

The role of the platform economy on technological factors. The first stage is to promote the marketization of the technological factor market and the deep integration of technological factors and rural markets. In the second stage, make technological inertia more significant, promote the deep integration of industry, academia, and research, and provide strong support for rural innovation. Create a market-demand-oriented technology research and development system, and achieve a better understanding of rural revitalization through data than people [17].

The role of the platform economy on labor factors. In the first stage, promote the provision of employment opportunities in rural industries and reduce the outflow of labor factors. In the second stage, new forms of labor, such as online labor, are emerging, helping urban labor to play a role in remote areas. User profiles predict the future labor market and reserve modern data talents for rural industrial development in advance.

**Table 1.** The role of platform economy on traditional production factors.

| Element/ Action Stage | Phase 1. Promotion Phase Solve Existing Problems | Phase 2: Leading Stage Promote and Lead the New Development of Traditional Production Factors |
|---|---|---|
| Technical elements | Promote the marketization of the technological factor market and promote the deep integration of technological factors and rural industrial markets | Make technological inertia more significant, promote the deep integration of industry, academia, and research, and provide strong support for rural industrial innovation; building a market-demand-oriented technology research and development system, achieving a better understanding of rural industries through data than people. |
| Capital factor | Improve the degree of capital factor market and alleviate the problems of unreasonable and uncoordinated capital allocation | Stimulate the potential of the capital market multiplier effect and guide the provision of support for the development of rural industries; Realize the dynamic adjustment of capital, predict the direction of the most Value investing in rural emerging areas, and guide the new flow of capital. |
| Labor factors | Promote the provision of employment opportunities in rural industries and reduce the outflow of labor factors | New forms of labor such as online labor are emerging, helping the urban labor force play a role in remote areas; user profiling predicts the future labor market and reserves modern data talents for rural industrial development in advance |
| Land elements | Revitalize potential land elements, monitor real-time planning policies related to land transfer and development, and promote the operation of the land market | Coordinate large-scale land use with small-scale land use to achieve optimal allocation of existing land resources; guide the development of land elements and prevent excessive capitalization of land elements |

The role of the platform economy on capital factors: in the first stage, improve the marketization of the capital factor market and alleviate the problems of irrational and uncoordinated capital allocation. In the second stage, stimulate the potential of the capital market multiplier effect and guide the provision of support for the development of rural industries. Realize the dynamic adjustment of capital, predict the most value investing direction of rural areas that are emerging, and guide the new flow of capital [11].

The role of the platform economy on land elements: in the first stage, activate the possible land elements, monitor the planning policies related to land circulation and development in real time, and promote the operation of the land market. In the second stage, coordinate large-scale land use with small-scale land use to achieve optimal allocation of existing land resources. Guide the development of land elements and prevent excessive capitalization of land elements. Based on the above analysis, hypothesis 3 is proposed.

**Hypothesis 3 (H3).** *The Platform economy produces a mesomeric effect through integration with technology, capital, labor, and land factors, thus indirectly promoting the integrated development of urban and rural areas.*

Based on the above assumptions, a conceptual model of the impact of the platform economy on urban and rural integrated development is proposed in Figure 1.

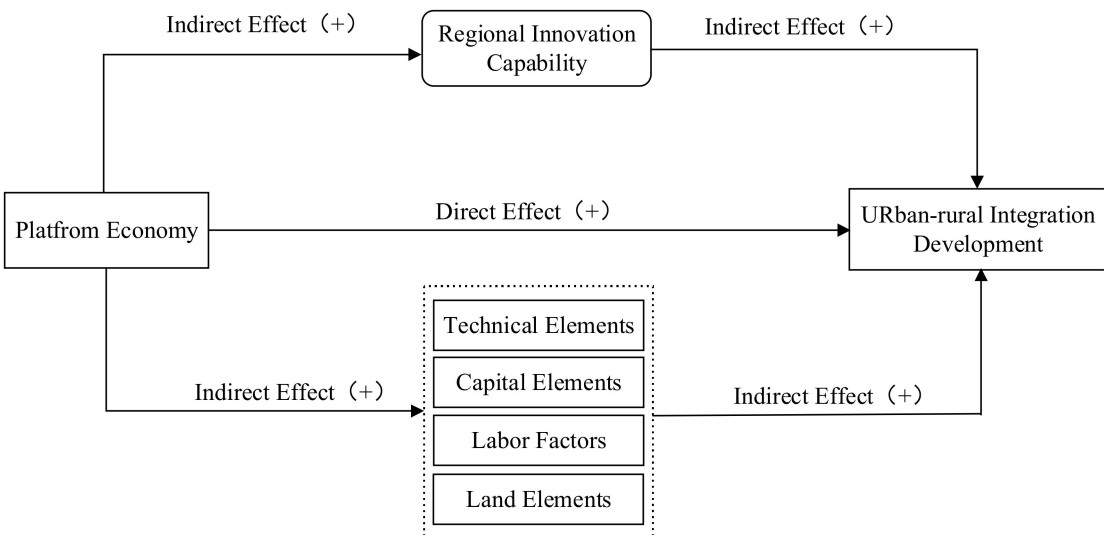

**Figure 1.** The conceptual model of the impact of the platform economy on urban and rural integrated development.

## 3. Methods and Data

### 3.1. Model Setting

To verify that the platform economy can directly promote the integrated development of urban and rural areas, this paper builds a basic model:

$$URI_{i,t} = \alpha_0 + \alpha_1 Platform_{i,t} + \alpha_c Control_{i,t} + \mu_i + \delta_t + \varepsilon_{i,t} \tag{1}$$

In Equation (1), $URI_{i,t}$ represents the level of urban–rural integration development in province $i$ during period $t$. $Platform_{i,t}$ represents the level of the platform economy in province $i$ during period $t$. $Control_{i,t}$ represents the set of other control variables that affect $URI_{i,t}$. $\mu_i$ and $\delta_t$ represent individual fixed effects and time fixed effects, respectively. $\varepsilon_{i,t}$ represents a random perturbation term.

In order to verify that the platform economy can indirectly promote urban and rural integrated development through regional innovation capability, this paper refers to the mesomeric effect model proposed by Chen and Wang [27]. On the basis of the basic model (1), the mesomeric effect model is further used to investigate the mesomeric effect of the platform economy on the integrated development of urban and rural areas. The mesomeric effect model is as follows:

$$URI_{i,t} = \beta_0 + \beta_1 Platform_{i,t} + \beta_c Control_{i,t} + \mu_i + \delta_t + \varepsilon_{i,t} \tag{2}$$

$$URI_{i,t} = \gamma_0 + \gamma_1 Platform_{i,t} + \gamma_2 Rip_{i,t} + \gamma_c Control_{i,t} + \mu_i + \delta_t + \varepsilon_{i,t} \tag{3}$$

The three-step test steps of the intermediary utility test are as follows: first, test the model (1). If the coefficient $\alpha_1$ is significantly positive, which indicates that the platform economy has a direct and significant role in promoting the development of urban and rural integration, then the next test can be carried out. Otherwise, it indicates that the mesomeric effect does not exist, and there is only a direct effect at this time. Secondly, on the basis of the first step, the model (2) is tested. If the coefficient $\beta_1$ is significantly positive, which means that the platform economy has a significant direct role in promoting regional innovation capability, then the next step can be taken, otherwise, stop the test and the mesomeric effect does not exist at this time. Finally, the model (3) is tested on the basis of the first and second steps. If both $\gamma_1$ and $\gamma_2$ coefficients are significantly positive, it indicates that the mediating variable has a certain degree of mesomeric effect and the test is over. Among them, $\alpha_1$ represents the total effect, $\gamma_1$ represents the direct effect, and $\beta_1 \times \gamma_2$ represents the mesomeric effect.

*3.2. Variable Selection*

1.   Explanatory variable: level of urban–rural integration development.

This article draws on the research method of Zhou et al. [28] and selects 18 indicators from five dimensions of urban–rural population integration, economic integration, social integration, spatial integration, and ecological integration, and constructs an evaluation index system for urban–rural integration development. The system includes comparative indicators that reflect the urban–rural gap, dynamic indicators that reflect the process of urban–rural integration, and state indicators that reflect the results of urban–rural integration. The weights of each indicator are calculated using the entropy method, as shown in Table 2.

**Table 2.** Evaluation index system for urban–rural integration development level.

| Dimension | Measurement Indicators | Indicator Measurement Formula | Properties |
|---|---|---|---|
| Population | Nonagricultural and Agricultural population ratio | Number of people in the second and third industries/Number of the first industry | + |
| | Population density | Urban population density/rural population density | − |
| | Urbanization | Urban population/ending permanent population | + |
| Economy | Per capita income ratio | Urban per capita income/rural per capita income | − |
| | Per capita out ratio | Urban per capita out/rural per capita out | − |
| | Proportion of nonagricultural | Value added of the secondary and tertiary sector/GDP | + |
| Society | Ratio of cultural, educational, and entertainment expenses | Urban cultural and educational expenditure/rural cultural and educational expenditure | − |
| | Healthcare expenditure ratio | Urban healthcare/rural healthcare expenditure | − |
| | Pension insurance rate | Urban and rural pension insurance expenditure/total expenditure of local finance | + |
| | Unemployment insurance rate | Number of people participating in unemployment insurance/ending permanent population | + |
| Space | Land allocation | Crop planting area/Built-up Area | + |
| | Traffic mobility | Highway and railway operating mileage/total administrative area | + |
| | Land urbanization | Built-up Area/total administrative area | + |
| | Passenger turnover rate | Passenger turnover/year-end resident population | + |
| Ecology | Greening level | Green coverage rate in built-up areas | + |
| | Domestic waste treatment | Harmless treatment rate of household waste | + |
| | Forest cover | Forest coverage/total land area | + |
| | Environmental protection | Environmental protection expenditure/total expenditure of local finance | + |

2.   Core explanatory variable: platform economy level.

The platform economy is a new economic form based on new infrastructure and modern information technologies such as the internet, the internet of things, and big data. It focuses on gathering resources, facilitating transactions, improving efficiency, building a platform industrial ecology, and promoting the efficient integration and innovative development of commodity production, circulation, and supporting services. We use the research of Ji et al. [29] to build an evaluation index system of the platform economy level from three dimensions of platform trading, platform infrastructure, platform products, and seven secondary indicators, as shown in Table 3.

**Table 3.** Evaluation index system of platform economy level.

| Dimension | Measurement Indicators | Indicator Measurement Formula | Properties |
|---|---|---|---|
| Platformized trading | Platform transaction level | E-commerce sales | + |
| | Platform procurement level | E-commerce procurement amount | + |
| | Platform scale | Number of e-commerce platform enterprises | + |
| Platformized infrastructure | Internet access situation | Number of broadband access ports per 100 people | + |
| | Internet resource situation | CN domain names per person | + |
| | Internet website points | Number of websites per 10,000 people | + |
| Platformized products | Platformized goods and services | Faster number of internet users per capita | + |

3. Mediating variables: regional innovation capability and traditional production factors.

Existing research on measuring regional innovation capabilities often uses patent applications, new product sales revenue, and patent authorizations to represent innovation levels. Due to the fact that many regions reward the number of patent applications as a measure of enterprise innovation performance, there may be some unqualified applications, leading to deviations in innovation measurement in some regions [30]. Generally, it is difficult to obtain new product-sales revenue data and there is a certain lag, so it is not adopted [31]. The number of patents granted is approved by the China National Intellectual Property Administration based on the creativity of enterprises applying for patents and includes invention, utility model, design, and other categories, which can reflect the level of R&D output [32]. This article uses the number of patent authorizations at each provincial level to represent regional innovation capabilities [33], denoted as *Rip*.

Traditional production factors include four types: technological factors, capital factors, labor factors, and land factors. Among them, the academic achievements of He and Yao are mainly used for reference to select the secondary indicators of technical factors, capital factors, and labor factors. Finally, the technology market turnover and high-tech industry R&D funds are used to measure the technical factors, human capital, and industrial fixed assets investment are used to measure the capital factors, and high-tech industry practitioners and informatization talents are used to measure the labor factors [34,35]. The land element is measured based on the academic achievements of Zhang et al., using the input amount of agricultural land and the proportion of cultivated land [36].

4. Control variables.

According to the specific situation of the explained variable, core explanatory variable, and intermediary variable, this article uses four variables as control variables: industrial structure (*Indus*), marketization level (*Urban*), population aging (*Odr*), and financial development level (*Finance*). In terms of detailed measurement, the industrial structure is measured by using the ratio of added value of the tertiary industry to that of the secondary industry based on the practices [37]; the marketization level is represented by the marketization process index based on the practices [38]; population aging refers to the practice [39], using the elderly dependency ratio as a measurement variable; the financial development level is measured by the ratio of the loan balance of financial institutions to the GDP of the current year with reference to Xu et al. thinking [40].

*3.3. Data Description*

Given the availability and comparability of data, Tibet and the Hong Kong, Macao, and Taiwan regions will not be considered for the time being. This paper selects the balanced panel data of 30 provinces (autonomous regions and municipalities) in China from 2011–2021 as the study sample. The data comes from the *China Statistical Yearbook* and China Tertiary sector of the economy *Statistical Yearbook*, which are downloaded from the

EPS data platform. Some missing values are filled by interpolation. Table 4 reports variable definitions and descriptive statistical results.

**Table 4.** Variable definition and descriptive statistics.

| Symbol | Name | N | Mean | Standard | Min | Max |
|:---:|:---:|:---:|:---:|:---:|:---:|:---:|
| *URI* | Level of urban–rural integration development | 330 | 0.284 | 0.096 | 0.126 | 0.797 |
| *Platform* | Platform economy level | 330 | 0.288 | 0.097 | 0.057 | 0.673 |
| *Rip* | Regional innovation capability | 330 | 0.313 | 0.124 | 0.077 | 0.882 |
| *Indus* | industrial structure | 330 | 1.226 | 0.685 | 0.428 | 6.497 |
| *Urban* | Marketization level | 330 | 5.878 | 2.124 | 2.731 | 12.45 |
| *Odr* | Population aging | 330 | 0.487 | 0.187 | 0.173 | 0.834 |
| *Finan* | Financial development level | 330 | 1.361 | 0.457 | 0.675 | 2.596 |

## 4. Empirical Analysis

### 4.1. Benchmark Regression

Both the Hausman test and the Wald test strongly reject the original hypothesis, indicating that the fixed effects model is significantly superior to the random effects model and mixed regression model. Therefore, this article selects the individual and time bidirectional fixed effects model for empirical analysis. Among them, model (1) is the test result of examining the direct impact of the platform economy on urban and rural integrated development without adding control variables. The regression coefficient is positive, and it passes the significance level test of 1%, which means that the platform economy has a significant role in promoting urban and rural integrated development. From model (2) to model (5), the test results were gradually added with control variables, and the regression coefficient remained significantly positive, which confirmed the validity of research hypothesis 1. However, the estimated coefficient has dropped from 0.315 in model (1) to 0.216 in model (5), indicating that the promotion effect of the platform economy on urban and rural integrated development is analyzed separately without considering the control variables, which to some extent magnifies the impact of the platform economy.

From the perspective of control variables, the regression coefficient of the industrial structure is 0.018, which is significant at the 1% confidence level, indicating that the industrial structure has a significant role in promoting the integrated development of urban and rural areas because the industry is an important carrier for the integrated development of urban and rural areas, and is also an important carrier for the platform economy to be deeply integrated. The leapfrog development of secondary industry is the ballast to improve economic quality and the level of residents' income, which can better narrow the regional gap and the urban–rural gap; The regression coefficient of the marketization level is 0.014, which is significant at the 1% confidence level, indicating that the marketization level can significantly promote urban–rural integration development because the market economy has mobilized all human and material resources, improved resource flow and allocation efficiency, accelerated the speed of economic development, brought great wealth to society, and, to some extent, accelerated the two-way flow of urban–rural factors. The regression coefficient of population aging is −0.023, which is significant at the 1% confidence level, indicating that population aging is not conducive to urban–rural integration and development, as the continuous intensification of population aging has changed the proportion relationship between labor and caregivers and, to some extent, expanded the income gap among residents. The regression coefficient of the financial development level is 0.006, which is significant at the 1% confidence level, indicating that the level of financial development promotes the integrated development of urban and rural areas. Finance plays an important role in the realization of common prosperity for all people but the modern financial system is not perfect. In particular, although the financial institutions mainly serve rural finance and small and microfinance, their own strength is insufficient The relatively backward technology leads to insufficient depth and coverage of

its services, resulting in a less significant effect on promoting urban–rural integration and development, as shown in Table 5.

**Table 5.** Direct impact.

| Variable | M(1) | M(2) | M(3) | M(4) | M(5) |
|---|---|---|---|---|---|
| *Platform* | 0.315 *** | 0.257 *** | 0.220 *** | 0.206 *** | 0.216 *** |
| | (0.006) | (0.012) | (0.015) | (0.018) | (0.018) |
| *Indus* | | 0.024 *** | 0.026 *** | 0.024 *** | 0.018 *** |
| | | (0.006) | (0.006) | (0.006) | (0.006) |
| *Urban* | | | 0.010 *** | 0.010 *** | 0.014 *** |
| | | | (0.002) | (0.002) | (0.002) |
| *Odr* | | | | 0.001 ** | −0.023 *** |
| | | | | (0.001) | (0.001) |
| *Finan* | | | | | 0.006 *** |
| | | | | | (0.007) |
| *_cons* | 0.161 *** | 0.146 *** | 0.092 *** | 0.082 *** | 0.096 *** |
| | (0.003) | (0.005) | (0.015) | (0.025) | (0.017) |
| year | Yes | Yes | Yes | Yes | Yes |
| zone | Yes | Yes | Yes | Yes | Yes |
| $N$ | 330 | 330 | 330 | 330 | 330 |
| $R^2$ | 0.983 | 0.979 | 0.877 | 0.974 | 0.799 |

Note: ** and *** represent significant confidence levels of 5% and 1%, respectively. Standard quantity statistics are shown in parentheses. The same below (Tables 6 and 7).

### 4.2. Indirect Impact Testing

After verifying that the platform economy helps to promote the integrated development of urban and rural areas, we need to further analyze its intermediary mechanism to clarify the mechanism of the two and provide a theoretical basis for the platform economy to promote the integrated development of urban and rural areas. Tables 6 and 7 report the test results of the intermediary impact of platform economy on the integrated development of urban and rural areas.

**Table 6.** The indirect impact of regional innovation capability.

| Variable | M(6) | M(7) |
|---|---|---|
| *Platform* | 0.163 *** | 0.185 *** |
| | (0.018) | (0.057) |
| *Rip* | | 0.279 *** |
| | | (0.020) |
| *Indus* | 0.012 *** | 0.032 *** |
| | (0.006) | (0.006) |
| *Urban* | 0.001 *** | 0.010 *** |
| | (0.003) | (0.002) |
| *Odr* | −0.001 | −0.002 ** |
| | (0.001) | (0.001) |
| *Finan* | 0.011 | 0.025 *** |
| | (0.007) | (0.006) |
| *_cons* | 0.211 *** | 0.029 *** |
| | (0.017) | (0.019) |
| $\beta_1 \times \gamma_2$ | 0.0450 | |
| year | Yes | Yes |
| zone | Yes | Yes |
| $N$ | 330 | 330 |
| $R^2$ | 0.779 | 0.919 |

**Table 7.** The indirect impact of traditional production factors.

| Variable | Technical Elements | | Capital Elements | | Labor Elements | | Land Elements | |
|---|---|---|---|---|---|---|---|---|
| | **M(8)** *Techf* | **M(9)** *URI* | **M(10)** *Capf* | **M(11)** *URI* | **M(12)** *Laborf* | **M(13)** *URI* | **M(14)** *Landf* | **M(15)** *URI* |
| *Platform* | 0.680 *** (0.043) | 0.426 *** (0.033) | 0.301 *** (0.010) | 0.491 *** (0.033) | 0.863 *** (0.060) | 0.285 *** (0.028) | 0.595 *** (0.028) | 0.145 *** (0.078) |
| *Techf* | | 0.121 *** (0.070) | | | | | | |
| *Capf* | | | | 0.150 *** (0.081) | | | | |
| *Laborf* | | | | | | 0.430 *** (0.006) | | |
| *Landf* | | | | | | | | 0.235 *** (0.055) |
| *Indus* | 0.012 *** (0.016) | 0.022 *** (0.016) | 0.062 *** (0.007) | 0.022 *** (0.004) | 0.051 *** (0.016) | 0.043 *** (0.009) | 0.022 *** (0.006) | 0.132 *** (0.003) |
| *Urban* | 0.001 *** (0.013) | 0.020 *** (0.012) | 0.012 *** (0.016) | 0.024 *** (0.005) | 0.042 *** (0.001) | 0.012 *** (0.005) | 0.002 *** (0.006) | 0.122 *** (0.007) |
| *Odr* | −0.001 (0.011) | −0.012 ** (0.011) | 0.032 *** (0.006) | 0.012 *** (0.012) | 0.052 *** (0.021) | 0.034 *** (0.023) | 0.004 *** (0.006) | 0.134 *** (0.013) |
| *Finan* | 0.021 (0.017) | 0.015 *** (0.016) | 0.013 *** (0.006) | 0.037 *** (0.065) | 0.022 *** (0.005) | 0.031 *** (0.007) | 0.112 *** (0.009) | 0.002 *** (0.010) |
| *_cons* | 0.201 *** (0.027) | 0.019 *** (0.009) | 0.024 *** (0.021) | 0.221 *** (0.036) | 0.040 *** (0.079) | 0.208 *** (0.050) | 0.115 *** (0.021) | 0.195 *** (0.036) |
| $\beta_1 \times \gamma_2$ | 0.0823 | | 0.0452 | | 0.2589 | | 0.1398 | |
| year | Yes | Yes | Yes | Yes | Yes | Yes | Yes | Yes |
| zone | Yes | Yes | Yes | Yes | Yes | Yes | Yes | Yes |
| *N* | 330 | 330 | 330 | 330 | 330 | 330 | 330 | 330 |
| $R^2$ | 0.789 | 0.909 | 0.876 | 0.975 | 0.879 | 0.965 | 0.932 | 0.981 |

### 4.2.1. The Mesomeric Effect of the Platform Economy on Urban–Rural Integration through Regional Innovation Capability

It can be seen from the test results of model (5) that the regression coefficient of platform economy to urban–rural integrated development is 0.216, which is significantly positive, indicating that there is a mesomeric effect, which meets the requirements of the next test; so, the next test is carried out. It can be seen from the test results of model (6) that the regression coefficient of the platform economy to rural revitalization is 0.283, which is significantly positive, meaning that platform economy has a significant impact on the intermediary variable regional innovation capability, which also meets the requirements for the next test; so, the next test can be carried out. According to the test results of model (7), the regression coefficients of the platform economy and regional innovation capability on urban–rural integrated development are 0.185 and 0.279, respectively, both of which are significantly positive, indicating that the intermediary variable of regional innovation capability has a certain degree of mesomeric effect and the three-step test of mesomeric effect has been completed. Finally, by combining the test results with the calculation formula of the mesomeric effect, we can see that the total effect coefficient of the platform economy on urban and rural integrated development is 0.216, the direct effect coefficient is 0.163, and the mesomeric effect coefficient is 0.045. That is to say, the mesomeric effect of the platform economy on urban–rural integrated development through regional innovation capability accounts for 20.83%, the direct effect of the platform economy on urban–rural integrated development accounts for 79.17%, and the direct effect of the platform economy on urban–rural integrated development is 3.62 times of the indirect effect; that is, the impact of the platform economy on urban–rural integrated development is mainly direct. Research hypothesis 2 is verified. From the perspective of control variables, although the regression coefficient of the impact of each control variable on the development of

urban–rural integration has changed and the plus–minus sign of the impact coefficient is always consistent with the basic regression, which proves the robustness of the conclusions drawn in this paper.

4.2.2. The Mesomeric Effect of Platform Economy on Urban–Rural Integration through Traditional Production Factors

Taking technological factors as an example, model 8 represents the mesomeric effect test of platform economy on technological factors, and model 9 represents the test of the impact of platform economy and technological factors on the integrated development of urban and rural areas. The results show that both model 8 and model 9 tests are significantly positive at the 1% significant level, which indicates that platform economy has a mesomeric effect; Further comparison shows that the coefficient decreases from 0.080 to 0.426 after adding technical factors, indicating that platform economy has some mesomeric effect on technical factors and the coefficient of mesomeric effect is 0.0823. Similarly, the capital factor, labor factor, and land factor all have a partial mesomeric effect and the mesomeric effect coefficients are 0.0452, 0.2589, and 0.1398, respectively. That is to say, the mesomeric effect of the platform economy has the law of increasing capital, technology, land, and labor factors. Among them, the mesomeric effect coefficient of the data element to labor factor is the largest, which indicates that the data element makes it easier to improve labor productivity.

*4.3. Robust Test*

1.  Time-phased estimation. The platform economy has obvious time-stage characteristics. This paper divides the inspection interval into two stages, 2011–2015 and 2016–2021, and estimates them, respectively. The results are shown in model (16) and model (17) in Table 8. It can be found that the direct effect of the two stages of platform economy on urban–rural integration development is significantly positive and it can be considered that the direct promotion effect of platform economy on urban–rural integration development is relatively stable. As for the previous stage, the regression coefficient is larger, which may be because the focus of the early development of the platform economy is to improve the digital infrastructure construction and the digital industrialization process is relatively fast, so the direct impact on the urban–rural integration development is larger. In the second stage, the depth and breadth of the integration of the platform economy and the real economy have been greatly improved but the direct role of promoting the urban–rural integration development is not as good as in the first stage;

2.  Replace the core explanatory variable. This paper measures the level of platform economy from the three dimensions of platform transactions, platform infrastructure, and platform products, which may have the problem of missing other key information of the platform economy, leading to the incomplete measurement results of the platform economy. Therefore, the robustness of the estimation results can be tested by replacing the core explanatory variables using the research results of other scholars. Therefore, this paper uses the financial inclusion index measured by Peking University as the core explanatory variable and strictly follows the mesomeric effect model test steps to test the indirect impact of the digital economy on common prosperity. The results are shown in Table 8: model (18), model (19), and model (20). It can be found that the platform economy still meets the test conditions of the mesomeric effect model for urban and rural integration development and, through calculation, the total effect coefficient is 0.203, the direct effect is 0.176, the mesomeric effect is 0.027, and the mesomeric effect accounts for 13.30;

3.  Delete four municipalities directly under the central government. The level of platform economy of municipalities directly under the central government is relatively in the leading position in the country and the inclusion of municipalities directly under the central government in the full sample may amplify the promotion effect of the

platform economy. Therefore, this article will remove municipalities directly under the central government from the sample data for threshold effect testing and the results are shown in model (21) and model (22) in Table 8 It can be found that the platform economy still has a mesomeric effect on land factors, indicating that the platform economy can promote the integrated development of urban and rural areas through traditional production factors, which again confirms the robustness and reliability of the test results.

**Table 8.** Robust test.

| Variable | M(16) | M(17) | M(18) | M(19) | M(20) | M(21) | M(22) |
|---|---|---|---|---|---|---|---|
| *Platform* | 0.347 *** | 0.212 *** | 0.203 *** | 0.147 *** | 0.176 *** | 0.495 *** | 0.155 *** |
| | (0.019) | (0.020) | (0.008) | (0.003) | (0.007) | (0.018) | (0.038) |
| *Rip* | | | | | 0.186 *** | | |
| | | | | | (0.020) | | |
| *Landf* | | | | | | | 0.335 *** |
| | | | | | | | (0.055) |
| *Indus* | 0.026 *** | 0.046 *** | 0.116 *** | 0.012 *** | 0.032 *** | 0.013 *** | 0.043 *** |
| | (0.007) | (0.006) | (0.026) | (0.002) | (0.006) | (0.014) | (0.007) |
| *Urban* | 0.010 *** | 0.021 *** | 0.016 *** | 0.011 *** | 0.014 *** | 0.011 *** | 0.022 *** |
| | (0.002) | (0.003) | (0.012) | (0.003) | (0.012) | (0.006) | (0.015) |
| *Odr* | −0.002 ** | −0.001 * | −0.003 *** | −0.001 ** | −0.002 ** | −0.003 *** | −0.013 *** |
| | (0.001) | (0.001) | (0.011) | (0.001) | (0.001) | (0.012) | (0.004) |
| *Finan* | 0.025 ** | 0.025 *** | −0.011 *** | 0.021 *** | −0.015 *** | −0.002 *** | 0.009 *** |
| | (0.011) | (0.009) | (0.002) | (0.017) | (0.006) | (0.002) | (0.006) |
| *_cons* | 0.081 *** | 0.030 *** | 0.054 *** | 0.201 *** | 0.019 *** | 0.014 *** | −0.064 *** |
| | (0.017) | (0.021) | (0.007) | (0.007) | (0.009) | (0.003) | (0.014) |
| $\beta_1 \times \gamma_2$ | | | | | 0.0273 | | 0.1658 |
| year | Yes | Yes | Yes | Yes | Yes | Yes | Yes |
| zone | Yes | Yes | Yes | Yes | Yes | Yes | Yes |
| N | 150 | 150 | 330 | 330 | 330 | 260 | 260 |
| $R^2$ | 0.8977 | 0.7542 | 0.6843 | 0.7932 | 0.8368 | 0.796 | 0.823 |

Note: *, ** and *** represent significant confidence levels of 10%, 5% and 1%, respectively. Standard quantity statistics are shown in parentheses.

## 5. Discussion

### 5.1. Evaluation of the Impact of Platform Economy on Urban–Rural Integration Development

There are three main ways for the platform economy to affect the integrated development of urban and rural areas. First, it can directly promote population integration, economic integration, social integration, spatial integration, and ecological integration. The second is to indirectly promote regional innovation capabilities. The third is to indirectly promote through transmission mechanisms by empowering traditional production factors such as technology, capital, labor, and land.

First, the direct effect of the platform economy on urban and rural integrated development. The construction of digital rural areas and the implementation of rural revitalization policies have guided numerous urban technical talents to integrate into and build rural areas, promoting the two-way flow of the urban and rural populations. The development of digital platforms based on digital technology has spawned new formats and provided more employment and entrepreneurial opportunities for rural residents. Farmers can promote characteristic agricultural products and develop rural tourism through online live streaming, which helps promote rural economic development and gradually promote the integration of urban and rural economies. The digitization of the social public-service system is conducive to addressing the urban–rural gap. For example, rural residents can express their demands through digital government platforms, share high-quality medical resources in the city through online consultation, and share first-class educational resources in the city through cloud-based online courses, which is conducive to achieving a balanced development of urban–rural public services. The platform economy uses digital platforms

to break through urban and rural spatial constraints through digital technologies, such as artificial intelligence, the internet, and big data, to achieve real-time information sharing between urban and rural areas, and to apply digital technologies to urban talent, technology, facilities and other resource elements in rural areas, and to implement remote control, which can weaken the geographical distance between urban and rural areas and promote urban and rural spatial integration. The integration of the platform economy and traditional industry and agriculture can change the production modes of traditional industries, help promote the digital, low-carbon, and green transformation of urban and rural industries and agriculture, improve the resource conversion rate and energy utilization rate, and reduce pollutant emissions.

Second, the platform economy promotes the integrated development of urban and rural areas through regional innovation capability. The platform economy can provide more convenience for enterprise innovation by improving the efficiency of resource allocation, saving innovation costs, and changing the way of thinking. The platform economy drives the development of new business models by integrating innovative elements and reshaping user behavior, thereby improving regional innovation capabilities. The open characteristics of the platform economy make the business network more open and the access to knowledge resources more diversified, thus promoting more opportunities for technology to be developed and utilized. The platform economy can break the space limitation and realize resource sharing. Innovative technology will drive the transformation and upgrading of traditional industries, expand their development scale, and generate a large number of employment opportunities. People will migrate to cities with higher economic levels in pursuit of better income, promoting population mobility and agglomeration. At the same time, the benefits generated by urban resource factors will feed back rural areas in other forms, promoting urban–rural interaction and integration. Innovative technology will change the extensive development model of rural agriculture, promote the recombination of production factors, improve agricultural production efficiency, drive rural economic growth, gradually narrow the urban–rural economic gap, and promote urban–rural integration development. High-tech industries are prone to form industrial clusters and can promote technology spillovers. At this time, the radiation and diffusion effects of urban innovative technologies can be fully utilized to promote the transformation and upgrading of rural industries in the region, effectively driving rural economic development, narrowing the urban–rural gap, and promoting urban–rural integration.

Finally, the platform economy plays an indirect role in the integrated development of urban and rural areas through traditional production factors. The platform economy can improve the two-way flow speed of traditional production factors and indirectly promote the integrated development of urban and rural areas. The role of the platform economy is to promote technological factors, promote the marketization of the technological factor market, promote the deep integration of technological factors and rural markets, make technological inertia more significant, promote the deep integration of industry, university, and research, and provide strong support for rural innovation. The role of the platform economy on labor factors promotes the provision of employment opportunities for rural industries, reduces the outflow of labor factors, and new forms of labor such as online labor have begun to emerge, helping the urban labor force play a role in different places. User portraits predict the future labor market and reserve data talents for rural industry development modernization in advance. The role of the platform economy on capital factors improves the degree of capital factor market, alleviates the problems of irrational and uncoordinated capital allocation, stimulates the potential of the capital-market multiplier effect, and guides the development of rural industries to provide support. The role of the platform economy on land factors invigorates possible land factors, monitors land circulation and development-related planning policies in real time, promotes land-market operation, coordinates large-scale land use and small-area land use, and achieves optimal allocation of existing land resources, guiding the development of land elements and preventing excessive capitalization of land elements.

### 5.2. Digital Twin Land

A digital twin is a new land in the platform economy and, also, a digital land in the digital era. This study finds that among the traditional factors of production, the mesomeric effect of the platform economy to promote the integrated development of urban and rural areas through land is greater than technology and capital, only second to labor, indicating that land is still the basis for farmers to survive in the era of the platform economy. However, although the emergence of digital twins has not changed the essence of land, it has changed the form of expression and existence of land, and digital land has burst into vitality.

Land is one of the most important inputs in the agricultural means of production. However, due to various reasons, there are problems of poor circulation and low utilization efficiency of land elements. The circulation of arable land is not smooth. There is a pair of contradictions in the development of rural industries, which are the lack of farmland and the lack of farmland for the tiller. The relatively low efficiency of the agricultural industry has led to farmers abandoning agriculture and turning to other industries, resulting in the phenomenon of abandonment. In addition, some farmers have neglected land management, resulting in dark wasteland, low land-use efficiency, and heavy losses in agricultural production. Although rural land transfer has improved the efficiency of farmland use, there are still problems with short terms and high rents in rural land transfer. This not only dampens farmers' enthusiasm for investing in land but also reduces agricultural returns and weakens the motivation of excellent workers to participate in modern agriculture. The decentralization and scale management of arable land resources are other major contradictions facing the revitalization of rural industries. Under the household responsibility system, cultivated land is evenly distributed according to the labor force or population. However, with the increase in population, the per capita arable land area is relatively small. With the shortage of land resources, the per capita arable land area will become smaller, and the revitalization of rural industries has put forward certain requirements for moderate-scale management. The utilization efficiency of homestead land is low. Due to the lack of long-term and detailed planning, as well as the lack of standardized channels to compensate for idle housing and rural housing relocated to cities, the current rough use of rural housing has led to the construction of new houses without demolishing old houses. Crowded homesteads, and the phenomenon of "hollowing out" is already very serious. If we consider the long-term idle of many houses on rural homesteads, the problem of low efficiency in rural land use becomes even more prominent. Digital land will overturn the traditional form of land existence and compensate for the drawbacks of poor farmland circulation, decentralized farmland, and the shortage of farmland. Digital twins can map objects that exist in reality to digital platforms and produce products that society needs in clusters.

### 5.3. Limitations and Future Directions

From a theoretical perspective, this study helps to reveal the relationship between the platform economy and urban–rural integrated development. Although the platform economy has emerged for many years, and China's e-commerce platform law has also ranked first in the world, the research on platform economy in narrowing the gap between urban and rural areas and promoting the integrated development of urban and rural areas is relatively scarce. There are few studies on the impact of platform economy on urban–rural integration development from the perspective of traditional production factors in the literature, especially on the new land of digital platforms. This paper studies and discusses the role of the platform economy in urban–rural integration development through the two transmission paths of regional innovation capability and traditional production factors. From the four traditional production factors of technology, capital, labor, and land, it is proposed that labor and land are the key factors affecting urban–rural integration development in the digital era, especially the generation of digital twin land.

This study also has certain limitations and requires further research. First, this article only proposes the concept of digital land from a theoretical perspective, defining it as a

fusion of traditional production factors such as land and digital platforms in the digital era, and conducting research on it. Second, due to space constraints, this paper does not study the heterogeneity of the impact of platform economy on urban–rural integrated development. Future research can be conducted on regional heterogeneity and multidimensional heterogeneity. Third, this study mentioned the comprehensive evaluation index system of the platform economy, which was not measured due to space constraints. Future research can specifically measure the level of the platform economy and its spatiotemporal differentiation characteristics.

## 6. Conclusions and Recommendations

Based on panel data of 30 provinces (autonomous regions, municipalities directly under the central government) in China from 2011 to 2021, this paper innovatively integrates platform economy, regional innovation capability, traditional production factors and urban–rural integrated development into a unified framework, proposes a conceptual model of platform economy on urban–rural integrated development, analyzes the theoretical mechanism of the platform economy's impact on urban–rural integrated development on the basis of existing literature, and empirically tests the direct effect of the platform economy on urban–rural integrated development mesomeric effect and transmission mechanism also test the robustness of the empirical test results. The results show that, first, the platform economy can indeed directly promote the development of urban–rural integration. The addition of spatial variables and robustness tests show that this result is robust, which fully shows that improving the level of the platform economy is conducive to narrowing the gap between urban and rural areas. Second, the platform economy can also indirectly have a significant positive impact on urban and rural integration development by promoting regional innovation capability; that is, the platform economy can indirectly promote urban and rural integration development through technological innovation and innovation-factor aggregation. Third, there is a transmission mechanism between the platform economy and traditional factors of production, and the mesomeric effect of the platform economy on technology, capital, land, and labor increases in turn. Fourth, digital land is a new form of land in the platform economy, and digital twins provide a carrier for digital land.

Based on the research findings of this article, the following four policy recommendations are proposed.

First, based on the basic fact that platform economy can directly promote the integrated development of urban and rural areas, accelerate the pace of digital industrialization and digital transformation of traditional industries, pay special attention to the high-quality development of platform economy, and improve the governance system of the platform economy. In recent years, China's platform economy has developed at an unprecedented speed, with a wide range of radiation and profound influence. The platform economy is turning to a new stage of deepening application, standardized development, and inclusive sharing. In addition, China has formed a strategic system of platform economy that is linked horizontally and linked vertically. In view of the brutal expansion of digital capital, cross-border monopoly of digital leading enterprises, big data killing, and algorithm collusion in recent years, we should start with upgrading governance tools, improving the governance system, and improving the ability to identify new monopolies to build a modern governance system for the platform economy.

Second, the platform economy has an indirect impact on urban–rural integration development by boosting regional innovation capability, which proves that regional innovation capability, as the intermediary between the platform economy and urban–rural integration development, can transmit the dividend of the platform economy to urban–rural integration development. Regional innovation capability is the fundamental path of integrated urban and rural development and, also, the key to solving the difficulties of achieving common prosperity for all people. This also determines the particularity of the realization mechanism of integrated urban and rural development in China. This particularity can be summarized as that in the "trinity" distribution mechanism; not only

the role of farmers' income should be reflected but also the role of agriculture and rural areas should be reflected, emphasizing its role in improving farmers' income levels and optimizing their income structure.

Third, the platform economy has an indirect effect on the integrated development of urban and rural areas through integration with traditional production factors, which proves that the platform economy has a promoting effect on traditional production factors. We should increase investment in rural digital infrastructure, enhance the digitization level of rural agriculture, accelerate the digitization speed of traditional production factors, encourage agricultural digital platforms to support rural agriculture, improve the efficiency of resource allocation in rural areas, fully utilize idle resources in rural areas, support the entry of agricultural digital platforms into rural agriculture, stimulate the integration of traditional production factors into digital platforms for digitization, and accelerate the two-way flow of traditional production factors.

**Funding:** This work was supported by the National Social Science Foundation Program of China (No.22CJY040).

**Data Availability Statement:** The experimental data is mainly downloaded from the EPS data platform. The website of this data platform is https://www.epsnet.com.cn/index.html#/Index, accessed on 24 June 2023. It was used to support the findings of this study are available from the corresponding author upon request.

**Conflicts of Interest:** The author declares that he has no conflict of interest.

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
