# Peer review of "The Impact of the Platform Economy on Urban–Rural Integration Development: Evidence from China"

_land, doi:10.3390/land12071417_

Round 1

Reviewer 1 Report

Dear Author,

thank you for interesting and sound research. However, I have a few suggestions to improve your paper.

1. Please review your abstract. Your paper is a way better than the abstract. Please avoid ";" mark as it is not used in a right way. 

2. I find the part 2 quite a problematic one. Sections 2.1, 2.2 and 2.3 are not referred to any scientific sources. Why do you say that "Platform economy helps to promote the integration of urban and rural population." Who says so? And there are many more such examples in this part. Please consider using scientific literature in this part.

3. Discussion and conclusions. What is the input of your paper to the science? In line 610 you are saying "from a theoretical perspective", please refer to the theory, please explain your contribution to the science, e.g., what have you confirmed as the previous researchers did, what have you found new, etc.

4. Thank you for a well written limitations and future research part. 

You are giving more recommendations than conclusions, therefore I suggest linking theoretical part to the results and to reveal how does your paper provide additional knowledge to the scientific part of the field.

There are some typos, but the quality of English is adequate. 

Author Response

My original paper ID is land-2496299- The Impact of Platform Economy on Urban-rural Integration Development: Evidence from China

I would like to thank all the reviewers for their suggestions. They are all very pertinent, which is very useful for me to improve the level of my paper. Here are some explanations of the reviewer's suggestions and questions.

  1. Please review your abstract. Your paper is a way better than the abstract. Please avoid ";" mark as it is not used in a right way.

Thanks for your proposal. Following your suggestion, I have rewritten the abstract.

Abstract: As a new economic system composed of data driven, platform supported and network coordinated economic activity units, platform economy provides new impetus for the integrated development of urban and rural areas. This paper uses China's provincial panel data from 2011-2021 to measure the development level of the platform economy and the urban-rural inte-gration development index using the entropy weight method and empirically tests the direct and indirect effects of the Platform economy on the urban-rural integration development. The re-search finds that: (1) Platform economy has a significant role in promoting the development of urban-rural integration. After introducing a series of robustness tests such as Instrumental vari-ables estimation; the results are still significant. (2) Regional innovation capability is an important transmission mechanism for platform economy to enable urban-rural integrated development. Platform economy can indirectly promote urban-rural integrated development by improving re-gional innovation capability. (3) There is a significant mesomeric effect between the traditional factors of production and the platform economy and the mesomeric effect of capital, technology, land , labor factors increases in turn. (4) The digital twin platform is a kind of digital land gener-ated in the platform economy and has become an important place and value source for products in the digital era.

  1. I find the part 2 quite a problematic one. Sections 2.1, 2.2 and 2.3 are not referred to any scientific sources. Why do you say that "Platform economy helps to promote the integration of urban and rural population." Who says so? And there are many more such examples in this part. Please consider using scientific literature in this part.

Thanks for your proposal. We have annotated the relative content according to your suggestion, enhancing the scientific nature of this section.

  1. Discussion and conclusions. What is the input of your paper to the science? In line 610 you are saying "from a theoretical perspective", please refer to the theory, please explain your contribution to the science, e.g., what have you confirmed as the previous researchers did, what have you found new, etc.

Thanks for your proposal. This is a very good suggestion. There are three main contributions:

Based on panel data of 30 provinces (autonomous regions, municipalities directly under the Central Government) in China from 2011 to 2021, this paper innovatively in-tegrates platform economy, regional innovation capability, traditional production fac-tors and urban-rural integrated development into a unified framework, proposes a conceptual model of platform economy on urban-rural integrated development, ana-lyzes the theoretical mechanism of platform economy's impact on urban-rural inte-grated development on the basis of existing literature, and empirically tests the direct effect of platform economy on urban-rural integrated development mesomeric effect and transmission mechanism also test the robustness of the empirical test results. The results show that: first, the platform economy can indeed directly promote the devel-opment of urban-rural integration. The addition of spatial variables and robustness tests show that this result is robust, which fully shows that improving the level of platform economy is conducive to narrowing the gap between urban and rural areas. Second, platform economy can also indirectly have a significant positive impact on ur-ban and rural integration development by promoting regional innovation capability, that is, platform economy can indirectly promote urban and rural integration devel-opment through technological innovation and innovation factor aggregation. Third, there is a transmission mechanism between the platform economy and traditional fac-tors of production, and the mesomeric effect of platform economy on technology, cap-ital, land and labor increases in turn. Fourth, digital land is a new form of land in the platform economy, and digital twins provide a carrier for digital land.

  1. Thank you for a well written limitations and future research part.

Thank you very much for your recognition.

Reviewer 2 Report

The article has a good perspective on the topic selection, however, there are  still two issues  need to be optimized.

(1) The Research Methods (3. Methods and Data) as well as  Results do not  support the Conclusions well, especially the content mentioned in the abstract. First of all, the research method of this study is relatively simple, and it is difficult to reflect the Mesomeric effect(line 18) described in the abstract of the article.  in other words, the finds mentioned in the abstract (3) and (4) are not clearly explained in the research method. in addition, the  abstract lacks specific data support (e,g. when referring to some words such as still significant , it is better to have a numbers to support that );

(2) The research hypothesis (2. Theoretical Hypothesis) lacks literature support. As the core content of this study, this section  of Line111-268  requires the support of some relevant literature.

In addition, some  results such as Empirical Analysis are best presented in graphical form and need to express the differences among provinces of China.

English Language is ok

Author Response

My original paper ID is land-2496299- The Impact of Platform Economy on Urban-rural Integration Development: Evidence from China

I would like to thank all the reviewers for their suggestions. They are all very pertinent, which is very useful for me to improve the level of my paper. Here are some explanations of the reviewer's suggestions and questions.

  1. The Research Methods (3. Methods and Data) as well as Results do not support the Conclusions well, especially the content mentioned in the abstract. First of all, the research method of this study is relatively simple, and it is difficult to reflect the Mesomeric effect(line 18) described in the abstract of the article.  in other words, the finds mentioned in the abstract (3) and (4) are not clearly explained in the research method. in addition, the  abstract lacks specific data support (e,g. when referring to some words such as still significant , it is better to have a numbers to support that );

Thanks for your proposal. We mainly use the Mesomeric effect model to test the Mesomeric effect of Platform economy on urban and rural integration through the intermediary variable of regional innovation capability. Model (1), model (2) and model (3) are the most commonly used Mesomeric effect models in academia. Its principle is mainly divided into three programs:

First, test the model (1). If the coefficient is significantly positive, which indicates that the platform economy has a direct and significant role in promoting the development of urban and rural integration, then the next test can be carried out. Otherwise, it indicates that the mesomeric effect does not exist, and there is only a direct effect at this time. Secondly, on the basis of the first step, the model (2) is tested. If the coefficient  is significantly positive, which means that the platform economy has a significant direct role in promoting regional innovation capability, then the next step can be taken, otherwise stop the test, and the mesomeric effect does not exist at this time. Finally, the model (3) is tested on the basis of the first and second steps. If both  and  coefficients are significantly positive, it indicates that the mediating variable has a certain degree of mesomeric effect, and the test is over. Among them,  represents the total effect,  represents the direct effect, and  represents the mesomeric effect.

  1. The research hypothesis (2. Theoretical Hypothesis) lacks literature support. As the core content of this study, this section of Line111-268 requires the support of some relevant literature.

Thanks for your proposal. We have added relevant literature and annotated it based on your suggestion.

Reviewer 3 Report

The subject is relevant and interesting. The structure of the model is clear, easy to construct and well described. The argumentation and methodology in the paper is convincing. I much like the scientific and logical layout and flow of this paper leading to the conclusions. The nature of this paper was such that it needs to be understandable and accessible both to academic audiences and policymakers. 

 The minor remarks are as follows:

Comment 1: Their driving effects have significant spatial heterogeneity. It is necessary to add some words to describe the underlying reasons for this phenomenon.

Comment 2: it is necessary to give a detailed reason Why did this study adopt these research methods?

Comment 3: I think the paper needs to be proofread by a native speaker.

Author Response

My original paper ID is land-2496299- The Impact of Platform Economy on Urban-rural Integration Development: Evidence from China

I would like to thank all the reviewers for their suggestions. They are all very pertinent, which is very useful for me to improve the level of my paper. Here are some explanations of the reviewer's suggestions and questions.

  1. Their driving effects have significant spatial heterogeneity. It is necessary to add some words to describe the underlying reasons for this phenomenon.

Thanks for your proposal. Based on your suggestion, we have added descriptive language.In fact, there should be some spatial heterogeneity among the 30 provinces, and there should be a trend of high in the east, low in the west, and flat in the middle. Both direct and indirect effects exhibit spatial heterogeneity, mainly due to the resource endowments between different provinces.

  1. It is necessary to give a detailed reason Why did this study adopt these research methods?

Thanks for your proposal.

To verify that the platform economy can directly promote the integrated development of urban and rural areas, this paper builds a basic model:

       (1)

In equation (1), represents the level of urban-rural integration development in province  during period , represents the level of platform economy in province  during period .  represents the set of other control variables that affect . and  represent individual fixed effects and time fixed effects, respectively. represents a random perturbation term.

In order to verify that platform economy can indirectly promote urban and rural integrated development through regional innovation capability, this paper refers to the mesomeric effect model proposed by Chen and Wang [27]. On the basis of the basic model (1), the mesomeric effect model is further used to investigate the mesomeric effect of platform economy on the integrated development of urban and rural areas. The mesomeric effect model is as follows:

         (2)

  (3)

The three-step test steps of intermediary utility test are as follows: First, test the model (1). If the coefficient is significantly positive, which indicates that the platform economy has a direct and significant role in promoting the development of urban and rural integration, then the next test can be carried out. Otherwise, it indicates that the mesomeric effect does not exist, and there is only a direct effect at this time. Secondly, on the basis of the first step, the model (2) is tested. If the coefficient  is significantly positive, which means that the platform economy has a significant direct role in promoting regional innovation capability, then the next step can be taken, otherwise stop the test, and the mesomeric effect does not exist at this time. Finally, the model (3) is tested on the basis of the first and second steps. If both  and  coefficients are significantly positive, it indicates that the mediating variable has a certain degree of mesomeric effect, and the test is over. Among them,  represents the total effect,  represents the direct effect, and  represents the mesomeric effect.

We mainly rely on reality, in addition to the direct effect of Platform economy on urban-rural integration, it can also have an indirect impact on urban-rural integration through traditional production factors and regional innovation capacity. So, based on existing theories, we make the following three assumptions:

H1. Platform economy can directly promote the integrated development of urban and rural areas.

H2. Platform economy can indirectly promote urban and rural integrated development through regional innovation capability.

H3. Platform economy produces mesomeric effect through integration with technology, capital, labor and land factors, thus indirectly promoting the integrated development of urban and rural areas.

  1. I think the paper needs to be proofread by a native speaker.

Thanks for your proposal. This is a very good suggestion. We have invited native English speakers to polish the article, and the grammar and vocabulary have been greatly improved.

Round 2

Reviewer 2 Report

Please revise and organize the references and clearly indicate which ones are in Chinese (in Chinese)

Author Response

My original paper ID is land-2496299- The Impact of Platform Economy on Urban-rural Integration Development: Evidence from China

I would like to thank all the reviewers for their suggestions. They are all very relevant, which is very useful for me to improve my paper level. Especially with this minor modification, I have benefited greatly. The following are some explanations of the reviewer's suggestions and questions.

  1. Please revise and organize the references and clearly indicate which ones are in Chinese (in Chinese)

Thanks for your proposal. I have indicated in red font in the references that those references are in Chinese. Thank you very much.

References

  1. Gagnon, J.; Goyal, S. Networks, markets, and inequality. American Economic Review 2017,107,1-30.
  2. Li, G .; Tao, T. E-commerce platform ecology and platform governance policy. Management World 2018,34,104-109.【中文参考文献】
  3. Glavas, C.; Mathews,S. How international entrepreneurship characteristics influence internet capabilities for the international business processes of the firm. International Business Review 2014,23,228-245.
  4. Seyoum, M.; Wu, R.; Yang, L. Technology spillovers from Chinese outward direct investment: The case of Ethiopia. China Economic Revies 2015,33:35-49.
  5. Cardona, M.; Kretschmer, T.; Strobel, T. ICT and productivity: conclusions from the empirical literature.Information Economics and Policy 2013, 25,109-125.
  6. Miyazaki, S.; Idota, H.; Miyoshi, H. Corporate productivity and the stages of ICT development. Information Technology and Management 2012,13,17-26.
  7. Mouelhi, R. B. A. Impact of the adoption of information and communication technologies on firm efficiency in the Tunisian manufacturing sector. Economic Modelling 2009,26,961-967.
  8. Forero, M. P. B. Mobile communication networks and Internet technologies as drivers of technical efficiency improvement. Information Economics and Policy 2013,25,126-141.
  9. Qu, J. B. Data Commodity and Capital Accumulation in Platform Economy. Finance & Economics 2020,390,40-49.
  10. Zhang, H. B.; Li. M. G. Factor Allocation and Governance Mode of Sharing Economy. Guizhou Social Sciences 2019,359,109-115. 【中文参考文献】
  11. Yi, X. R.; Chen. Y. Y.; Yu. W. A Study of the Essence and Operating Mechanism of Platform Economy. Jiangsu Social Sciences 2020,313,70-78+242. 【中文参考文献】
  12. Fan, Y. X.; Research on the Fiscal Policy Orientation for Urban-Rural Integrated Development in the Context of the Digital Economy. Frontiers 2021,210,52-58. 【中文参考文献】
  13. Chen, X, X.; Duan, B. Has the digital Economy narrowed the gap between urban and rural areas?—Empirical test based on mediating effect model.World Regional Studies 2022,31,280-291. 【中文参考文献】
  14. Pisano, P.; Pironti, M.; Rieple, A .Identify Innovative Business Models: Can Innovative Business Models Enable Players to React to Ongoing or Unpredictable Trends?.Entrepreneurship Research Journal 2015, 5,181-199.
  15. Armstrong, M. Competition in two-sided markets. The RAND Journal of Economics 2006,37,668-691.
  16. Drahokoupil, J.; Jepsen M .The Digital Economy and Its Implications for Labour: 1. The Platform Economy.Social Science Electronic Publishing 2017, 23,103–119.
  17. Paunov, C.; Rollo, V. Has the Internet Fostered Inclusive Innovation in the Developing World?.World Development 2016,78:587-609.
  18. Li, X. Z.; Li, J. Y. Research on the Influence of Digital economy Development on Urban-rural Income Gap. Journal of Agrotechnical Economics 2022,322,77-93. 【中文参考文献】
  19. Tang, H. T.; Xie, T. Digital Economy and Dual-Increasing of Farmers’ Income and Consumption. Journal of South China Agricultural University 2022,21,70-81. 【中文参考文献】
  20. Paul, J.; Prof, G. P.; Thompson , P.; Williams R. Are UK SMEs with active websites more likely to achieve both innovation and growth? .Journal of Small Business and Enterprise Development 2013, 20,934-965.
  21. Wei, J. Y.; Hu, R. Z.; Chen, Y. E. How does the Development of Digital Economy Affect the Urban-Rural Consumption Gap: Expanding or Narrowing?.Consumer Economics 2022,38,40-51. 【中文参考文献】
  22. He, L. H.; Wang, F.; Wang, C. M. How will the Digital Economy Drive China’s Rural Revitalization?.Inquiry into Economic Issues 2022,477,1-18. 【中文参考文献】
  23. Xie, F. S.; Wu ,Y.; Wang, S. S. A Political Economy Analysis of the Globalization of Platform Economics. Social Sciences in China 2019,288,62-81+200. 【中文参考文献】
  24. Synyutka, N.; Lutsyk, A .Transformation of Fiscal Policy and Fiscal Space under Conditions of Digital Technologies Expansion. Accounting and Finance 2018,4,108-113.
  25. Xie, L.; Han, W. L.Theoretical Logic and Practical Path of Digital Technology and Digital Economy to Promote Urban-rural Integration Development. Issues in Agricultural Economy 2022,515,96-105. 【中文参考文献】
  26. Zhou, D.; Qi, J. L.; Zhong, W. Y. Review of Urban-rural integration Evaluation: Connotation Identification, theoretical analysis, and system reconstruction . Journal of Natural Resources 2021,36,2634-2651. 【中文参考文献】
  27. Chen, Y.; Wang, Z. The Impact of Land Transfers on the Adoption of New Varieties: Evidence from Micro-Survey Data in Shaanxi Province, China. Land 2023, 12,632-655.
  28. Zhou, J. N.; Qin, F. Q.; Liu, J. Measurement, spatial-teporal evolution and influencing mechanism of urban-rural integration level in china from a multidimensional perspective. China Population Resources and Environment 2019,29,166-176. 【中文参考文献】
  29. Ji, Y. Y.; Zhang, M. X.; Feng, S. H. Research on the impact of platform economy on the upgrading of industrial structure- Based on the perspective of consumption platform. Systems Engineering- Theory & Practice 2022,42,1579-1590. 【中文参考文献】
  30. Zhang, J.;Gao, D. B.;Xia, Y. L.Do Patents Drive Economic Growth in China- An Explanation Based on Government Patent Subsidy Policy. China Industrial Economics 2016,334,83-98. 【中文参考文献】
  31. Cheng, G. B.; Ji, Y. Can improving innovation capabilities enhance urban economic resilience? .Modern Economic Research 2022,482,1-11+32. 【中文参考文献】
  32. Yu, D. J.; Jin, X. R. Comparative Study on the Regional Innovation Efficiency of Innovation Subjects. Science Research Management 2014,35,51-57. 【中文参考文献】
  33. Jinag, Q.;Li, J. Z.;Jiang, C. M. Can Platform economy improve regional innovation capability. Journal of Shandong University of Finance and Economics 2023,35,56-67. 【中文参考文献】
  34. He, X. J.;Wei, G. D.; Zhang, T. T. Evaluation and Empirical Study on the Degree of Collaborative Development between Regional Factor Endowment and Manufacturing Industry. China Soft Science 2016,12,163-171. 【中文参考文献】
  35. Yao, F. Research on the Impact of Factor Endowment on the Growth of Factor Intensive Manufacturing Industry: Based on the Analysis of Mechanical and Electronic Manufacturing Industries in Various Provinces and Regions. Inquiry into Economic Issues 2016,3,30-41. 【中文参考文献】
  36. Zhang, Z.; Tang Z.; Fan, B. Q. Research on land factor promoting economic growth under the background of factor marketization reform: based on the synergistic relationship between land factor and labor factor. China Soft Science 2023,387,192-202. 【中文参考文献】
  37. Larose, R .; Gregg, J. L. ; Strover , S. Closing the rural broadband gap: Promoting adoption of the Internet in rural America. Telecommunications Policy 2015, 31,359-373.
  38. Xiang, Y.; Lu, Q.; Li, Z. X.. Digital Economy Development Empowers Common Prosperity: Impact Effects and Mechanism. Securities Market Herald 2022,358,2-13.
  39. He, L.; Wang, F.; Wang, C. How will the digital economy drive china’s rural revitalization?. Inquiry into Economic Issues 2022,477,1-18. 【中文参考文献】
  40. Xu ,P .J.;Yang, H. L; Wei Q. analysis on the influencing factors of china’s common prosperity-based on the perspective of modern industrial system and consumption. Reform of Economic System 2022,234,16-24. 【中文参考文献】
